# Structural and quantitative alterations of gut microbiota in experimental small bowel obstruction

**Jiali Mo**[1☯], **Lei Gao**[1☯], **Nan Zhang**[2]*, **Jiliang Xie**[2], **Donghua Li**[3], **Tao Shan**[2], **Liuyang Fan**[1]

**1** Graduate school of Tianjin Medical University, Tianjin, China, **2** Department of Gastrointestinal Surgery, Nankai Hospital, Tianjin, China, **3** Department of Pharmacology, Tianjin Nankai Hospital, Tianjin, China

☯ These authors contributed equally to this work.
* zhangnannk@163.com

## Abstract

### Objective

To investigate structural and quantitative alterations of gut microbiota in an experimental model of small bowel obstruction.

### Method

A rat model of small bowel obstruction was established by using a polyvinyl chloride ring surgically placed surrounding the terminal ileum. The alterations of gut microbiota were studied after intestinal obstruction. Intraluminal fecal samples proximal to the obstruction were collected at different time points (24, 48 and 72 hours after obstruction) and analyzed by 16s rDNA high-throughput sequencing technology and quantitative PCR (qPCR) for target bacterial groups. Furthermore, intestinal claudin-1 mRNA expression was examined by real-time polymerase chain reaction analysis, and serum sIgA, IFABP and TFF3 levels were determined by enzyme-linked immunosorbent assay.

### Results

Small bowel obstruction led to significant bacterial overgrowth and profound alterations in gut microbiota composition and diversity. At the phylum level, the 16S rDNA sequences showed a marked decrease in the relative abundance of Firmicutes and increased abundance of Proteobacteria, Verrucomicrobia and Bacteroidetes. The qPCR analysis showed the absolute quantity of total bacteria increased significantly within 24 hours but did not change distinctly from 24 to 72 hours. Further indicators of intestinal mucosa damage and were observed as claudin-1 gene expression, sIgA and TFF3 levels decreased and IFABP level increased with prolonged obstruction.

### Conclusion

Small bowel obstruction can cause significant structural and quantitative alterations of gut microbiota and induce disruption of gut mucosa barrier.

**Data Availability Statement:** All relevant data are within the paper and its Supporting Information files.

**Funding:** This work was supported by the Tianjin Medical University under Grant [number

11601502-XK0135]. The funders had no role in study design, data collection and analysis, decision to publish, or preparation of the manuscript.

**Competing interests:** The authors have declared that no competing interests exist.

## Introduction

Small bowel obstruction (SBO) is a common indication in the emergency department and a common cause of hospitalization and surgical intervention globally [1]. SBO occurs when the passage of intestinal contents is interrupted, which is characterized by abdominal pain, vomiting, abdominal distention, and absolute constipation. A variety of pathologic processes can cause SBO, but in industrialized countries, adhesions are the most common etiology [2]. Other etiologies include incarcerated hernias, obstructive lesions (malignant and benign), and many infrequent reasons such as volvulus, intussusception, diverticulitis, and foreign bodies (including gallstones) [3]. Small bowel obstructions carry significant morbidity and mortality. In the UK, SBO accounted for as many as 51% of all emergency surgical admissions and the mortality ranges from 2% to 8% [4, 5]. Similar results have been found in the USA, emergency general surgeries related to SBO counted for 80% of morbidity and death in the emergency department, and the disease cost over $2 billion in inpatient annually due to a long period of ward and ICU stay [6].

In recent years, gut microbiota, also known as intestinal microbiota or intestinal flora, has turned out to be a new frontier in the study of human diseases. It has been estimated that the microbes in our gastrointestinal tract harbor over $10^{14}$ bacterial cells, tenfold higher numbers of human cells, which encode 150-fold more unique genes than the human genome [7]. Considerable evidence has proved that gut microbiota is a significant component in the maintenance of health and is associated with a large number of human diseases. Clinical and animal studies have suggested that diseases such as metabolic diseases [8], autoimmune disorders [9], neuropsychiatric diseases [10] and cancer [11] are closely related to dysbiosis of the gut microbiota. Dysbiosis often refers to an imbalance in the gut microbial community including changes in absolute quantity of community members or relative abundance of microbes [12].

Small bowel obstruction leads to a series of local and systemic consequences. Gut microbiota dysbiosis is an important change of both local and systemic pathological changes in SBO. Overgrowth of bacteria especially pathogenic ones may cause the impairment of the gut mucosa barrier and mucosal inflammation, as a result of mucosal disruption. This intestinal dysbacteriosis and damaged mucosa contribute to increased bacterial translocation. Translocated bacteria and endotoxins are responsible for the evolution of systemic inflammatory response syndrome and multiple organ dysfunction in SBO [13]. SBO can result in significant alterations in gut microbiota, however, most published research on this topic has focused on bacterial translocation, with little focus on structural and quantitative alterations of gut microbiota during the process of disease.

The determination of qualitative and quantitative changes of bacteria in SBO in previous studies basically adopted culture-dependent techniques [14–16]. However, most of the species in the human intestine are anaerobic and therefore are difficult to culture [17]. The introduction of high-throughput DNA sequence analysis has led to uncovering the composition and diversity of the gut microbiota. In the present study, 16S rRNA sequencing was used to investigate structural alterations of intestinal microbiota before and after obstruction in small bowels, and the dynamic changes of microbiota as the disease progressed. Sequencing consequences of entire microbial communities can only provide information of relative abundance of microbes [18]. Therefore, a quantitative PCR (qPCR) assay using genus-specific 16S rDNA primers was applied in the determination of quantitative alterations of gut microbiota in healthy rats and those with SBO, targeting the genus showing dramatic changes in proportions.

Besides, we also investigated the pathological changes in intestinal mucosal barriers. Further indicators were observed as claudin-1 gene expression representing tight junction barrier and secretory immunoglobulin A (sIgA) levels for the mucosal immune system, as well as

intestinal fatty acid binding protein (IFABP) and Trefoil factor 3 (TFF3) for mucosal damage. Finally, the correlation of the genus associated with disease and gut mucosal barriers were analyzed. The interactions of specific microbiota with these barriers contribute to explain how commensal and pathogenic microorganisms influence their hosts in SBO and their crucial roles of disease progression.

## Materials and methods

### Establishment of small bowel obstruction in rats

Male Wistar rats (n = 32) weighing between 220 and 260 g were used in the study and purchased from Beijing HFK Bioscience Co. Ltd. (Beijing, China). These animals were housed at 20–22˚C with a relative humidity of 50±10% in a light/dark cycle of 12/12 hours and allowed free access to food and water. The animals were housed in individual ventilated cages and provided with autoclaved chow and sterilized water. The experimental procedures were approved by Ethics Committee of Tianjin Nankai Hospital.

Thirty-two rats were randomly divided into four experimental groups (n = 8 per group): Sham (sham operation group), S1 (small bowel obstruction for 24 hours), S2 (small bowel obstruction for 48 hours), S3 (small bowel obstruction for 72 hours). All experiments were carried out under strictly aseptic conditions. Animals were intraperitoneally anesthetized with 1% sodium pentobarbital (40 mg/kg) and laparotomy was performed. Proximal colon and ileum with the cecum were taken out with micro-forceps and surgical scissors. A small loop of ileum with mesentery approximately 1 cm proximal to the ileocecal valve were selected. The mesentery was carefully incised parallel to the ileum to create a small window. In the animals with obstruction, an autoclaved flexible PVC tube (10 mm in length, 4 mm exterior diameter, 3 mm interior diameter) was cut longitudinally to open the tubing and formed a ring structure [19]. One end of the opened ring was inserted through the mesenteric window and brought into contact with the other. The ring was returned to a completed ring shape surrounding the ileum and closed with a suture. Then the intestines were placed back in the intraperitoneal cavity gently, followed by closure of the abdomen. The size of ring was sufficient to create a complete small bowel obstruction and could avoid focal necrosis of intestine compared to the ligation with silk suture. The control group underwent a similar procedure, but no obstruction ring used in these animals. The experimental animals were sacrificed 24, 48 and 72 hours after obstruction.

### Fecal sample collection and DNA extraction

Before surgery, feces of all the animals were collected for DNA isolation. Five rats of each group were randomly selected for fecal bacterial DNA extraction after operation. The content of the gut above the obstruction point was collected for analysis and the content of the same segment was collected in rats in the control group. Rats were anesthetized by intraperitoneal injection (40 mg/kg) of 1% sodium pentobarbital, and fecal contents above the obstruction point were collected aseptically and placed in sterile containers, then immediately frozen at -80˚C for further analysis. The rats were killed by cervical dislocation after samples were collected. For cervical dislocation, the animals were restrained by their heads and pulled on the tail until the crack sound indicating separation of cervical tissues. This procedure was performed by trained researchers using appropriate method. Fecal bacterial DNA was extracted using an EasyPure Stool Genomic DNA Kit (TransGen Biotech, China) according to the manufacturer's guidelines. The final DNA concentration and purification were measured by Nano-Drop 2000 UV-vis spectrophotometer (Thermo Scientific, Wilmington, USA), while the quality of the DNA was checked by 1% agarose gel electrophoresis.

## Histological evaluations

A 2 cm long ileal segment 2 cm proximal to the obstruction point was cut and rinsed with pre-cooled 0.9% Normal Saline and immersed in 10% formalin solution for 48 hours and dehydrated in a graded series of ethanol before embedded in paraffin. Serial sections were stained with Hematoxylin and Eosin (H&E). Each slide was selected five random visual fields and histological intestinal mucosal damage was assessed using Chiu's criteria (Table 1).

## Illumina sequencing of 16S rRNA genes

The extracted DNA was amplified using 16S rRNA V3-V4 region primers (338F: 5'– ACTC CTACGGGAGGCAGCAG–3' and 806R: 5'– GGACTACHVGGGTWTCTAAT–3') by PCR. Sequencing was performed with an Illumina MiSeq instrument and data were utilized to evaluate bacterial communities in fecal samples. The raw fastq files were demultiplexed and quality-filtered using QIIME and USEARCH. Operational Taxonomic Units (OTUs) were clustered with 97% similarity cutoff using the Greengenes reference database. The phylogenetic affiliation of each 16S rRNA gene sequences was analyzed by RDP Classifier (RDP Release 11) against the Silva (SSU123) 16S rRNA database using a confidence threshold of 70% [20]. Alpha diversity analysis (Observed species, Chao1, Shannon indexes and Coverage index) and beta diversity analysis (PCA and PCoA) were determined using QIIME software and R packages. Alpha diversity was used for analyzing microbial richness of the samples while beta diversity was for similarities between samples. The relative abundance percentage in different taxonomic levels were calculated and presented as the histogram graph and heat map. Linear discriminant analysis (LDA) effect size (LEfSe) was performed to identify bacterial taxa differentially represented among all groups, as well as between each two selected groups.

## Enumeration of target bacterial groups by quantitative PCR

Based on the sequencing results, the quantitative PCR (qPCR) assays for selected bacterial groups (total bacteria, *Romboutsia* spp., *Turicibacter* spp., *Lactobacillus* spp., *Akkermansia* spp., *Escherichia coli*, *and Bacteroides* spp.) were performed to determine the quantitative alterations of total bacteria and specific bacterial genera in SBO. Sequences of the corresponding primers used for qPCR are shown in Table 2. Fecal bacterial DNA (100 ng) was amplified in a 50 μl reaction buffer containing 2×TransTaq-T PCR Supermix (TransGen Biotech, China) and 10 pmol/μl specific primers. The PCR conditions were as below: 94˚C for 5 min, followed by 35 cycles at 94˚C for 30 s, annealing for 30 s at respective $T_m$ temperature (Table 2), and 72˚C for 60 s. Melting curve analysis was carried out at the end.

The standard curves for target bacterial groups quantification were generated by serial dilution of plasmid DNA containing the 16S rRNA target sequences. Copy numbers for standard curves were calculated based on the following formula [21]: copies/μl = (NL×A×$10^{-9}$)/(660×n), where NL was the Avogadro constant ($6.02×10^{23}$ molecules per mole), A was the

**Table 1. Criteria for Chiu's intestinal mucosal damage.**

| Score | Grading | Criteria |
|-------|---------|----------|
| 1 | Grade 0 | Normal villi |
| 2 | Grade 1 | Submucosal space on top of villi, capillary congestion |
| 3 | Grade 2 | Expanded submucosal space, isolation of intestinal mucosa and submucosa |
| 4 | Grade 3 | Isolation of intestinal mucosa and submucosa was extended to bilateral villi |
| 5 | Grade 4 | Blunted villi, exposure of lamina propria and vessels, inflammatory infiltration |
| 6 | Grade 5 | Digestion and disintegration of lamina propria, with bleeding and ulceration |

**Table 2. qPCR primer information and predicted product size.**

| Target bacterial group | Primer sequences (5'-3') | Predicted PCR product size (bp) | Annealing temp (˚C) | Reference |
|---|---|---|---|---|
| Total bacteria | UniF: GTGCTGCATGGTCGTCGTCA | 148 | 60 | [22] |
| | UniR: ACGTCGTCCACACCTTCCTC | | | |
| *Romboutsia* spp. | F: TGACATCCTTTTGACCTCTC | 282 | 54 | [23] |
| | R: GCCTCACGACTTGGCTG | | | |
| *Lactobacillus* spp. | F: AGCAGTAGGGAATCTTCCA | 341 | 58 | [24] |
| | R: CACCGCTACACATGGAG | | | |
| *Turicibacter* spp. | F: CAGACGGGGACAACGATTGGA | 141 | 63 | [25] |
| | R: TACGCATCGTCGCCTTGGTA | | | |
| *Akkermansia* spp. | F: CAGCACGTGAAGGTGGGGAC | 329 | 60 | [26] |
| | R: CCTTGCGGTTGGCTTCAGAT | | | |
| *Escherichia coli* | F: CATGCCGCGTGTATGAAGAA | 96 | 57.5 | [27] |
| | R: CGGGTAACGTCAATGAGCAAA | | | |
| *Bacteroides* spp. | F: CTGAACCAGCCAAGTAGCG | 230 | 53 | [28] |
| | R: CCGCAAACTTTCACAACTGACTTA | | | |

molecular weight of standard DNA molecules (ng) and n was the length of amplicon (bp). The copy numbers of target bacterial 16S rRNA genes per milligram of sample was calculated using the following equation as previously reported [21]: copy numbers/mg = (QM × C × DV)/ (S × W), where QM was the quantitative mean of the copy number, C was the DNA concentration of each sample (ng/μl), DV was the dilution volume of extracted DNA (μl), S was the DNA amount subjected to analysis (ng) and W was the sample weight subjected to DNA extraction (mg). copy numbers = copy numbers/mg × total volume of fecal contents.

## Intestinal claudin-1 mRNA expression

Expression of claudin-1 mRNA was examined by real-time PCR. Total RNA was extracted from intestinal segments (50 mg) from the ileum (4 cm distal to the obstruction) using TRIzol (Invitrogen, USA) reagent by following the manufacturer's instructions. First-strand cDNA was synthesized from 1 μg mRNA using a TransScript Fly First-Strand cDNA Synthesis Super-Mix kit (TransGen Biotech, China). The primers sequences were as follows: Claudin1 (forward: 5'-CTGTCGTTGGTGCTGTTTCA-3', reverse: 5'-CCTCTGCCCAACCTCAGTAA-3'); GAPDH (forward: 5'-TGCACCACCAACTGCTTAGC-3', reverse: 5'- GGCATGGACT GTGGTCATGAG-3'). Real-time PCR was performed with an Applied Biosystems 7500 FAST system using SYBR Green. The mRNA expression was analyzed by $2^{-\Delta\Delta Ct}$.

## Serum sIgA, IFABP and TFF3 levels

Blood (5 ml) was collected from the abdominal aorta in sterile tubes and centrifuged at 3000 rpm for 15 minutes. Serum was collected and stored at -20˚C. Levels of serum sIgA were measured by enzyme-linked immunosorbent assay, using Rat Secretory Immunoglobulin A ELISA kit (Catalog: E02S0003, BlueGene, Shanghai, China) following the manufacturer's instructions.

Levels of serum IFABP and TFF3 were determined using Rat Intestinal fatty acid binding protein ELISA kit (Catalog: E02I0340, BlueGene, Shanghai, China) and Rat Trefoil Factor ELISA kit (Catalog: E02T0132, BlueGene, Shanghai, China) according to the manufacturer's instructions and the results were calculated using standard curve.

## Statistical analysis

Data were presented as mean ± standard deviation. Differences between groups were analyzed by one-way ANOVA followed by post-hoc LSD tests. Pearson's correlation analysis was used. $P<0.05$ was considered statistically significant. Analysis was performed using IBM SPSS Statistics version 24 (IBM SPSS, Chicago, Illinois, USA).

## Results

### Modified rat model of small bowel obstruction

A complete small bowel obstruction was established in the rat by using the flexible PVC ring surgically placed surrounding the terminal ileum. The complete SBO model was successfully created when rats did not produce fecal pellets and developed visibly distended abdomens. Body weight, food intake, water intake and intestinal diameter were recorded. The general condition of the rats is shown in Fig 1. All animals in the Sham group were in good condition. Rats in SBO groups showed manifestations in just 24 hours including fatigue, weight loss, reduced food and water intake. The symptoms were aggravated rapidly with prolonged obstruction duration.

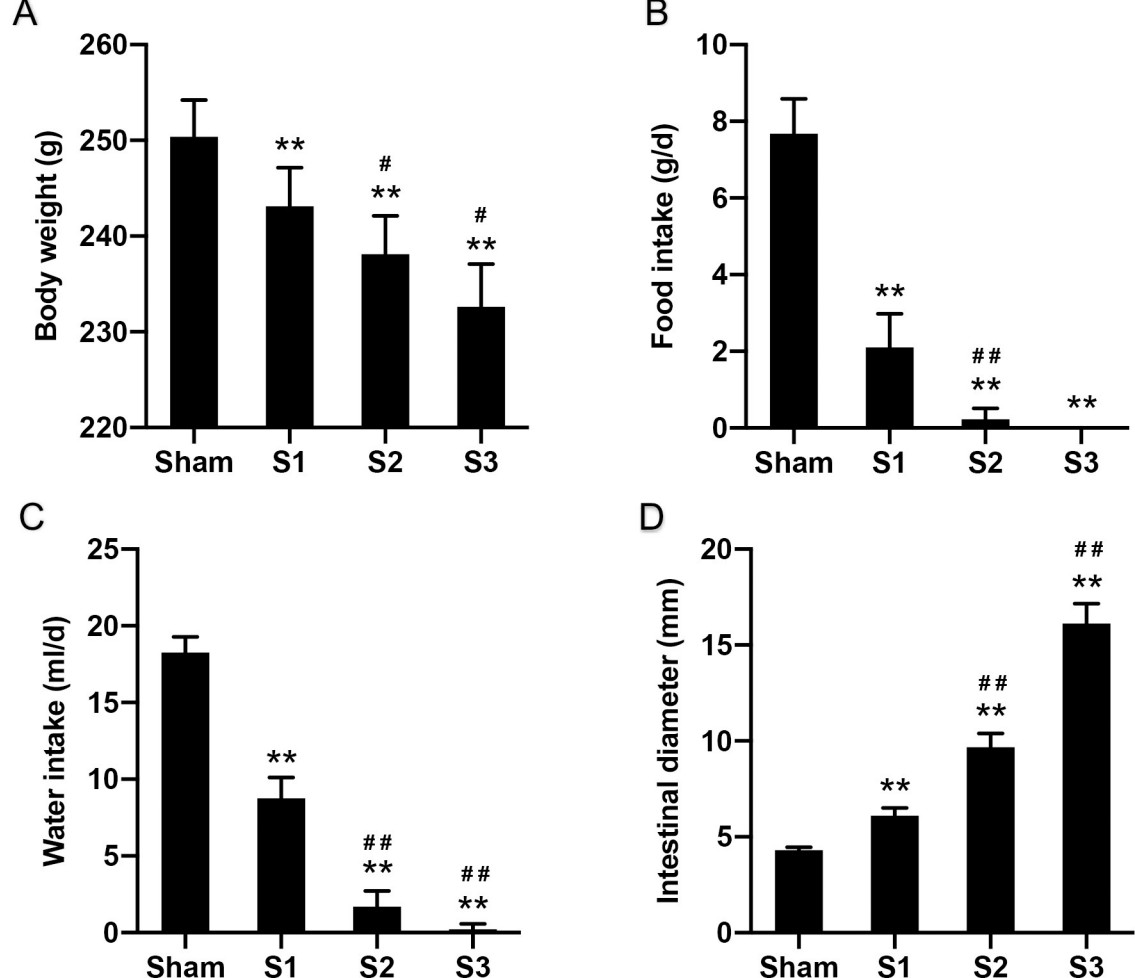

**Fig 1. The general condition of the rats.** Body weight (A), food intake (B), water intake (C), intestinal diameter (D) of the rats. Data are presented as the mean ± SD, n = 5. * P<0.05, ** P<0.01 vs. Sham group; #P<0.05, ## P<0.01 vs. the preceding group.

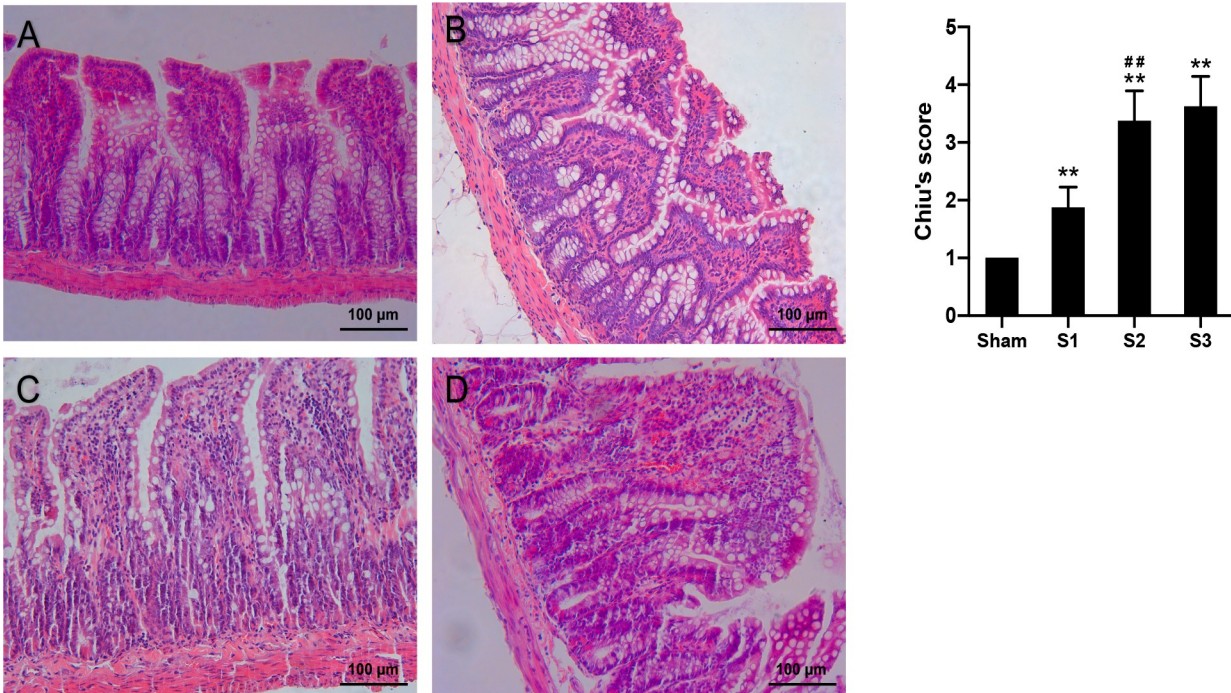

**Fig 2. Histological micrographs of intestinal tissue sections and intestinal mucosal damage scores.** Intestinal sections were assessed by H&E staining (A-D). Intestinal mucosal injury was quantified by Chiu's scores (H). Data are presented as the mean ± SD, n = 5. * P<0.05, ** P<0.01 vs. Sham group; #P<0.05, ## P<0.01 vs. the preceding group.

The appearance of the intestines and abdominal cavity was observed after laparotomy. In the Sham group, intestines appeared normal in diameter, color and mobility. In the S1 group, small intestines at the proximal section of the obstruction were dilated mildly and in hyperperistalsis, and gas-liquid accumulation occurred. In the S2 group, intestines were significantly dilated and in poor peristalsis, and there was significant gas-liquid accumulation in the intestine and some exudate in the abdominal cavity. In the S3 group, small intestines showed aggravated dilation, along with a large amount of gas-liquid accumulation. The diameter of intestine was expanded to more than three times its normal size.

The histological features in ileal tissues are presented in Fig 2. In the Sham group, the ileum mucosal layers were arranged in order and the structure of the villi was normal. In the S1 group, mild derangement of intestinal mucosa was observed, with slightly uneven villus heights and partial widening, mucosa edema and inflammatory cell infiltration. In the S2 group, there were significant damages of the small intestinal mucosa, with partial villous atrophy, separation of submucosa and muscular layer and capillary congestion in the villi. The S3 group showed partial villous fusion and epithelium shedding, with inflammatory cells infiltration and focal hemorrhage. There were fibrous tissue proliferation, vasodilation and hyperemia in sub-mucosa, as well as serositis and mesenteritis (Fig 2A–2D). Chiu's intestinal mucosal damage scores were significantly higher (p < 0.01) in the SBO groups compared to sham-operated group and gradually increased with prolonged obstruction duration (Fig 2E).

**Microbiota diversity and composition in obstruction.** The results of the principal coordinate analysis (PCoA) before surgery showed that all samples were clustered (Fig 3), presenting all the four groups had similar microbiota composition before experiment.

A total of 20 DNA samples from 5 sham and 15 SBO rats were available for the sequencing analysis. The alpha diversity parameters of the samples are displayed in Fig 4. Observed

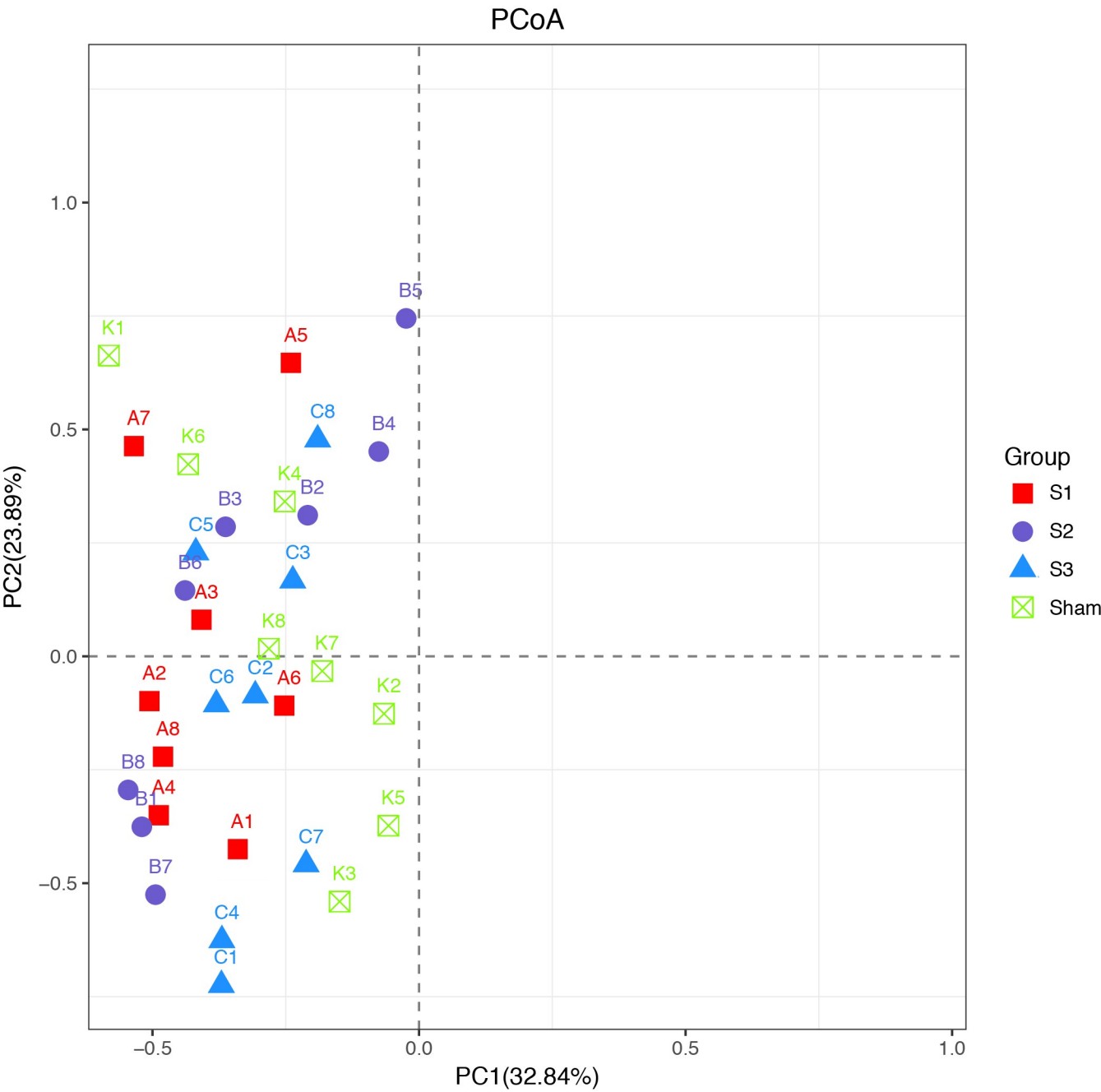

**Fig 3. Principal coordinate analysis (PCoA) analysis before surgery.**

species, Chao1, Shannon index values in SBO groups were significantly higher than in Sham and gradually increased with prolonged obstruction duration, indicating a greater microbiota diversity in blocked intestine. The coverage index ranged from 95% to 100% and showed a decrease in values after obstruction, suggesting that the sequence numbers of each sample were high enough to capture the majority of the known bacterial taxa and more unidentified gut bacteria appeared in SBO groups.

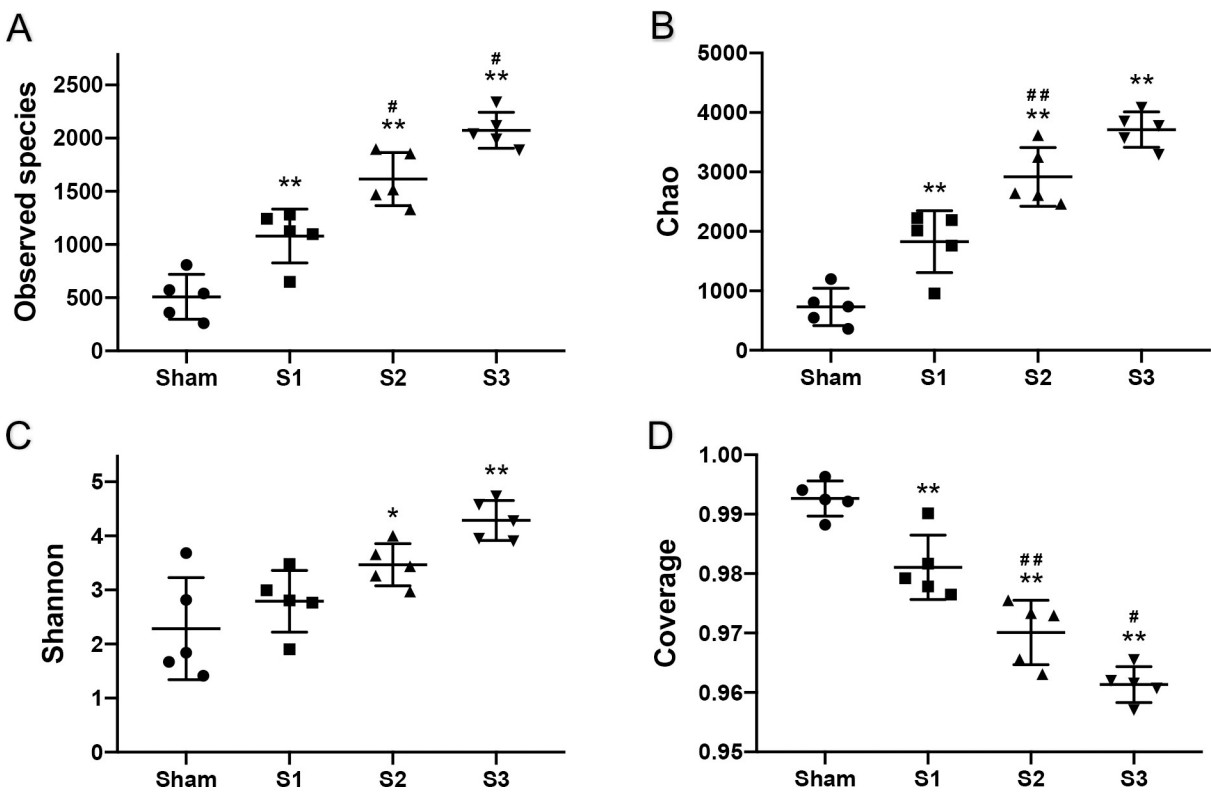

**Fig 4. Alpha diversity of fecal contents from sham and obstruction rats at different time points.** Observed species (A), Chao1 (B), Shannon (C), Coverage (D) of the gut microbiomes. The data are presented as the mean ± SD, n = 5. * P<0.05, ** P<0.01 vs. Sham group; #P<0.05, ## P<0.01 vs. the preceding group.

The results of principal component analyses (PCA) are shown in Fig 5A. Principal component 1 and principal component 2 respectively explained 58.14% and 14.01% of the variance of variables, and the cumulative proportions were up to 72.15%. PCA results showed that sham-operated group formed a cluster distinct from SBO groups, indicating the differences in bacterial community composition between normal rats and SBO rats were significant. Interestingly, samples in S1, S2, and S3 groups were close to each other and got closer to sham-operated group with prolonged obstruction duration. The principal coordinate analysis (PCoA) based on unweighted UniFrac distance matrix were performed to view and compare phylogenetic information of the bacterial communities among the samples. The results of the PCoA showed that samples in each group were clustered (Fig 5B). There were significant differences in structures of bacterial communities not only between the sham operation group and SBO groups, but also among rats in S1, S2 and S3 groups.

The distribution of the bacterial population is shown in Fig 6. Small bowel obstruction resulted in considerable microbial communities shift at different taxonomical levels. At the phylum level, there were significant decreases in the relative abundance of Firmicutes ($p < 0.01$) and Actinobacteria ($p < 0.05$) in SBO compared to sham. Meanwhile, SBO caused significant increases ($p < 0.01$) in the relative abundance of Proteobacteria, Verrucomicrobia and Bacteroidetes (Fig 7A). However, the relative abundance of Firmicutes gradually increased with prolonged obstruction while Proteobacteria and Verrucomicrobia showed a decreasing trend. In addition, the abundance of Bacteroidetes continued to increase after obstruction and became dominant phylum together with Proteobacteria and Verrucomicrobia in SBO as time

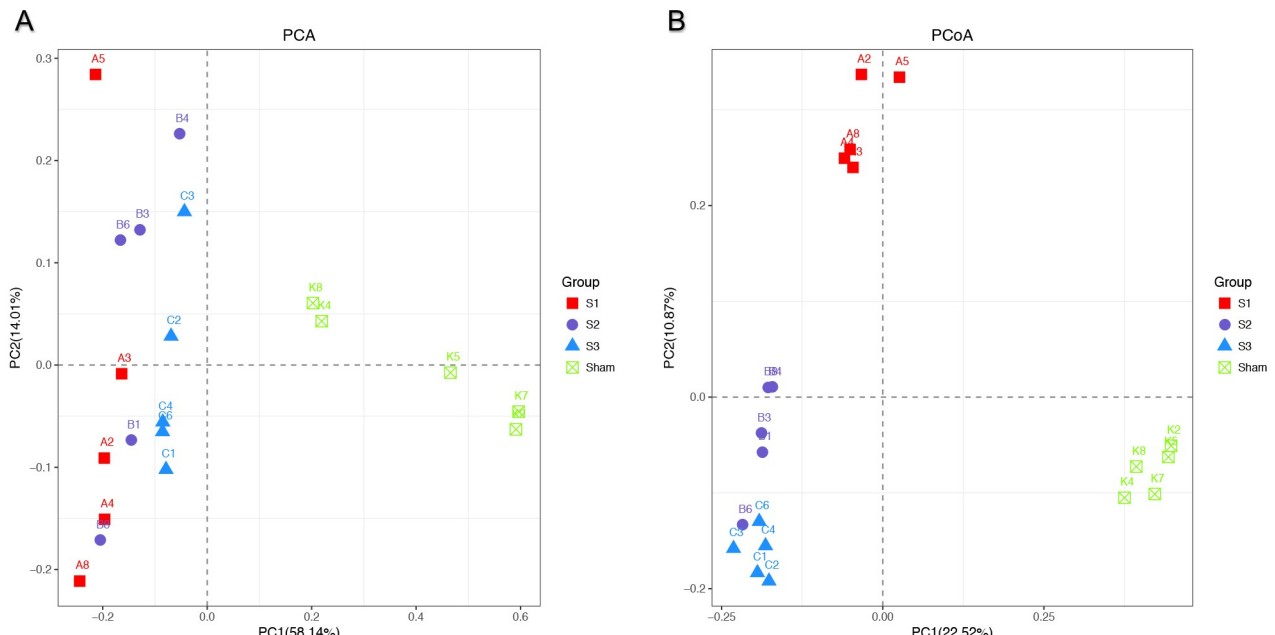

**Fig 5.** The plots of principal component analysis (PCA) (A) and principal coordinate analysis (PCoA) based on unweighted UniFrac distance metric (B). Samples of rats were presented by different color-filled symbols. Analysis of similarity revealed clustering among rats with obstruction and sham-operated rats.

progressed (Fig 7A). At the genus level, the abundance of *Romboutsia*, *Turicibacter*, *Lactobacillus* and *Peptoclostridium* decreased ($p < 0.01$) in SBO, whereas *Akkermansia*, *Escherichia-Shigella*, *Bacteroides*, *Enterobacter* and *Lysinibacillus* increased significantly ($p < 0.01$) and showed similar changing trends as the phylum level that they belonged to (Fig 7B). The heatmap showed differences and time-dependent changes in relative abundances and compositions of the microbial communities among the SBO and sham groups at the genus taxonomic level (Fig 8).

LEfSe analysis was carried out to identify the differentially abundant features in SBO. The results showed that S1 group was characterized by higher abundance of *Akkermansia* (belonging to Verrucomicrobia phylum) and *Escherichia-Shigella* (belonging to Proteobacteria). S2 group was marked by a higher level of *Bacteroides* (belonging to Bacteroidetes). In S3 group, the gut bacteria were characterized by an obviously enhance in *Bacteroides*, *Lysinibacillus* (belonging to Firmicutes), *Enterobacter* (belonging to Proteobacteria). Microbiota in sham control samples had higher levels of the Firmicutes phylum ($p < 0.05$, LDA score > 4.0) (Fig 9).

## qPCR analysis of target bacterial groups

Total volume of fecal contents above the obstruction point was recorded (Fig 10A). The volume of contents at 72 h was tenfold higher numbers of sham-operated rats. The copy numbers ($\log_{10}$ transformation) of 16S rRNA gene of total bacteria and six bacterial genera were compared among the sham group and different time points in SBO groups (Fig 10B–10H). Total bacteria, *Akkermansia* spp., *Escherichia coli*, *and Bacteroides* spp. were significantly higher ($p < 0.01$) in the diseased compared to sham-operated rats. In contrast, *Romboutsia* spp. and *Turicibacter* spp. were significantly lower ($p < 0.01$) in the disease groups. The amount of *Lactobacillus* spp. remained at roughly the same level as sham-operated rats at early stage of obstruction. The changing trends in the absolute bacterial cell numbers of specific bacterial

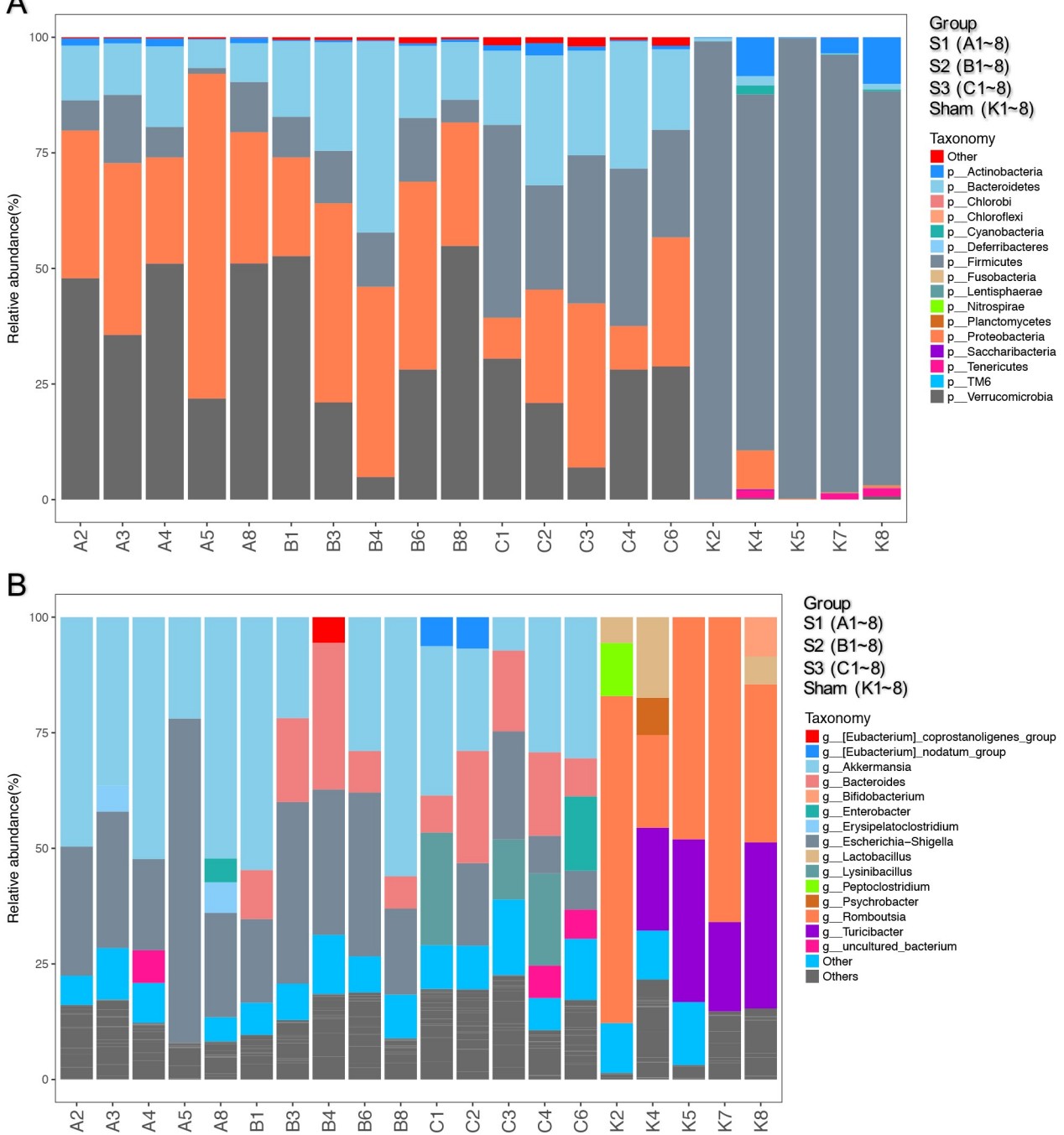

**Fig 6. Comparison of small intestinal microbiota composition in sham and SBO rats.** Intestinal microbiota composition at Phylum level (A), Genus level (B) are shown in the figure.

genera were not completely consistent with the relative abundance of these genera from 24 to 72 hours.

## Associations between gut microbiota and intestinal mucosal barriers

There were no statistically significant differences in the expression of claudin-1 mRNA between Sham and S1 groups, but it decreased significantly at 48 h of obstruction (P<0.01).

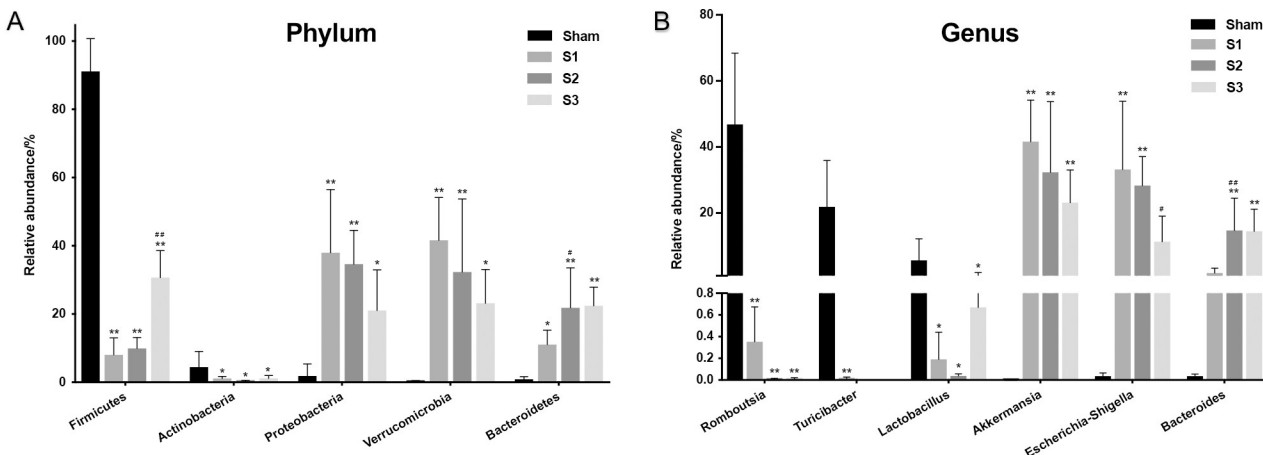

**Fig 7. Comparison of relative abundances of the bacterial taxa exhibiting significant changes in sham and SBO rats.** Relative abundances of the bacteria at Phylum level (A), Genus level (B) are shown in the figure. Data are presented as the mean ± SD, n = 5. * P<0.05, ** P<0.01 vs. Sham group; #P<0.05, ## P<0.01 vs. the preceding group.

Expression continued to decline within 72 h after obstruction, indicating severe damage in the intestinal epithelium (Fig 11A). Furthermore, sIgA levels, dominating humoral immunity at the intestinal mucosa, were significantly decreased with prolonged obstruction time (Fig 11B). Serum IFABP levels were significantly increased with prolonged obstruction time, starting at 24 hours (Fig 11C), while TFF3 levels decreased sharply from 24 hours and they decreased more significantly at day 2 (Fig 11D).

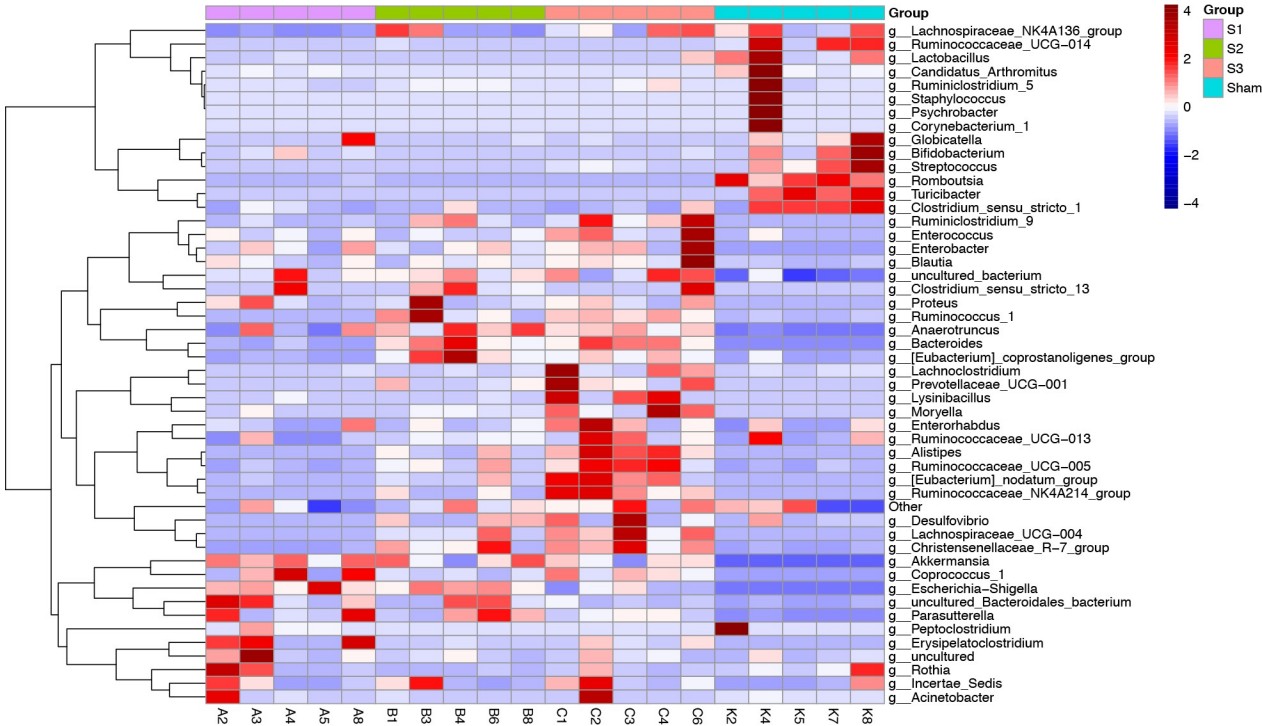

**Fig 8. Heatmap analysis at the genus level among four groups.**

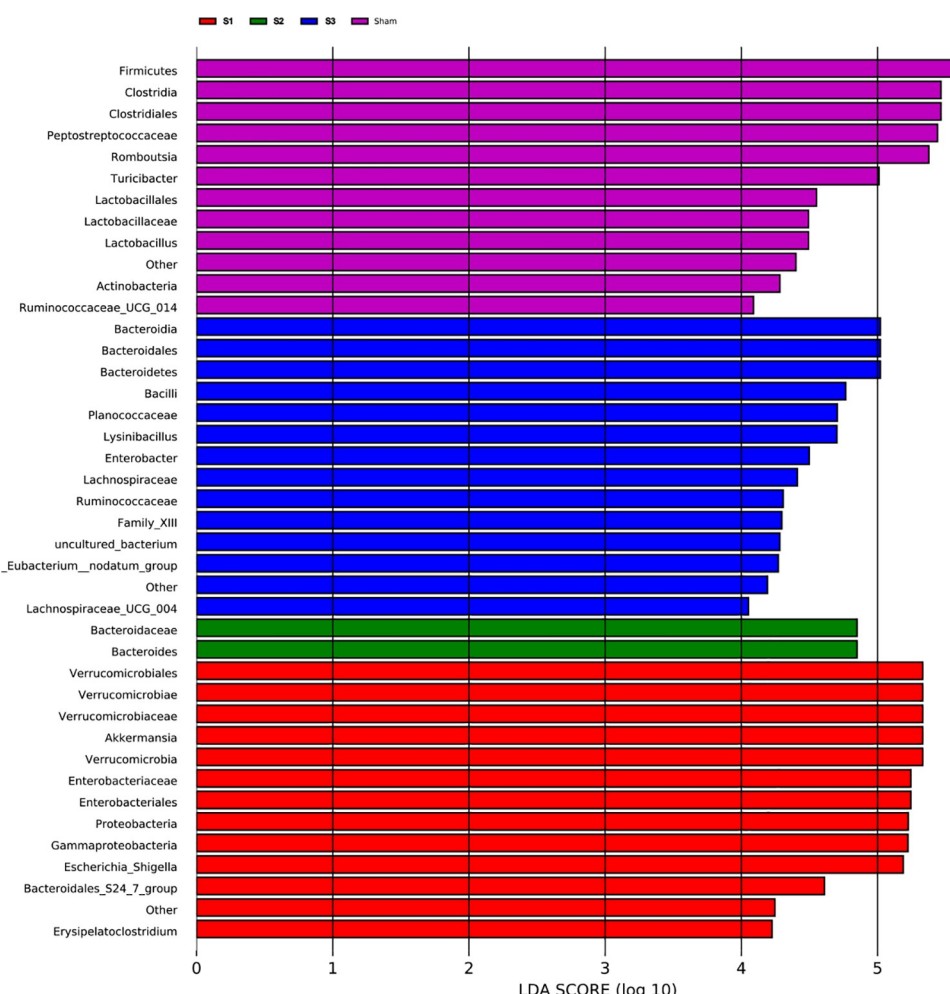

**Fig 9. LEfSe analysis for identification of differentially abundant taxa in SBO compared to sham rats.** The length of the horizontal bar represents the LDA score in log scale, and only taxa meeting an LDA score threshold > 4.0 are listed.

Lastly, we assessed the correlations between target bacterial genera and indicators of intestinal mucosal barrier function. Claudin-1 mRNA displayed positive correlation with *Romboutsia* spp. (P<0.05), while showed negative correlation with *Akkermansia* spp., *Escherichia coli*, and *Bacteroides* spp. (P<0.05). Positive correlations of sIgA were observed with *Romboutsia* spp. and *Turicibacter* spp. (P<0.01), whereas negative correlation with *Akkermansia* spp., *Escherichia coli*, and *Bacteroides* spp. (P<0.01). IFABP showed negative correlation with *Romboutsia* spp. and *Turicibacter* spp. (P<0.01), while displayed positive correlation with *Bacteroides* spp. (P<0.01). Positive correlations of TFF3 were observed with *Romboutsia* spp. and *Turicibacter* spp. (P<0.01), whereas negative correlation with *Bacteroides* spp. (P<0.01) (Table 3).

## Discussion

Small bowel obstruction is a common indication for surgical procedures and hospital admission globally, which is associated with high morbidity and hospitalization costs [3, 29]. Acute SBO can lead to a series of local and systemic pathophysiological derangements. Previous

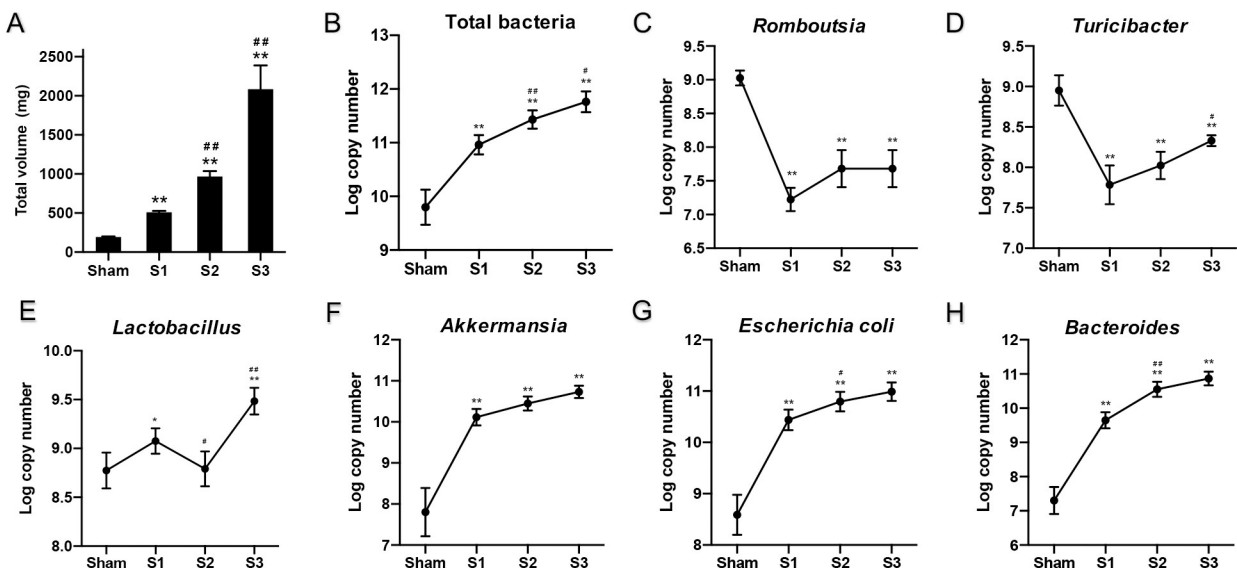

**Fig 10. Total volume of fecal contents and changes in quantification of target bacterial groups with prolonged obstruction time.** Data are presented as the mean ± SD, n = 5. * P<0.05, ** P<0.01 vs. Sham group; #P<0.05, ## P<0.01 vs. the preceding group.

research has suggested that the hypersecretion in obstructed bowel is caused by the overgrowth of intestinal bacteria, because in the germ-free dogs, no hypersecretion happens in the intestinal lumen and no serious morphological changes occur in villus structure [30]. Besides, translocated bacteria and endotoxins are responsible for the systemic pathophysiological consequences. Therefore, intestinal bacteria may play an important role in the pathogenesis of SBO. The analysis based on 16S rRNA gene sequencing and qPCR data provided a basic overview of the composition and quantification of the gut microbiota differ between healthy control rats and rats with SBO.

In this study, we established a complete small bowel obstruction animal model using flexible PVC rings with the optimal size. Most complete bowel obstruction models were usually caused by ligating the intestines with silk threads [31, 32]. However, these methods are unstable in practice since tightness and intensity of ligature cannot be controlled precisely. We used flexible PVC tube with a specific size to form an obstruction ring (10 mm in length, 4 mm exterior diameter, 3 mm interior diameter) after a series of preliminary experiments. Research has shown that the diameter of the ileum of rats weighted 200-300g is about 4.3±0.3mm [33], so the ring was 1–2 mm smaller than the intestinal luminal diameter. Compared to the silk ligation, this ring was sufficient to create a complete bowel obstruction while protecting intestine against focal necrosis.

To the best of our knowledge, there are few studies that focus on the bacterial contents of the obstructed small bowel and the results are almost all based on culture-dependent techniques. Our study is the first to report qualitative and quantitative changes of bacteria before and after small bowel obstruction, we use high-throughput DNA sequence analysis and quantitative PCR to determinate gut microbiota alterations during the process of disease. In previous studies, *E. coli* has been proved to play a predominant role in small bowel obstruction since 1960 [15, 16]. Another study showed that the amount of *Bacillus bifidus* gradually decreased and *Enterobacteriaceae* increased after obstruction, presenting an inverted ratio of these two types of bacteria as time progressed [14]. Our results are in agreement with these previous studies.

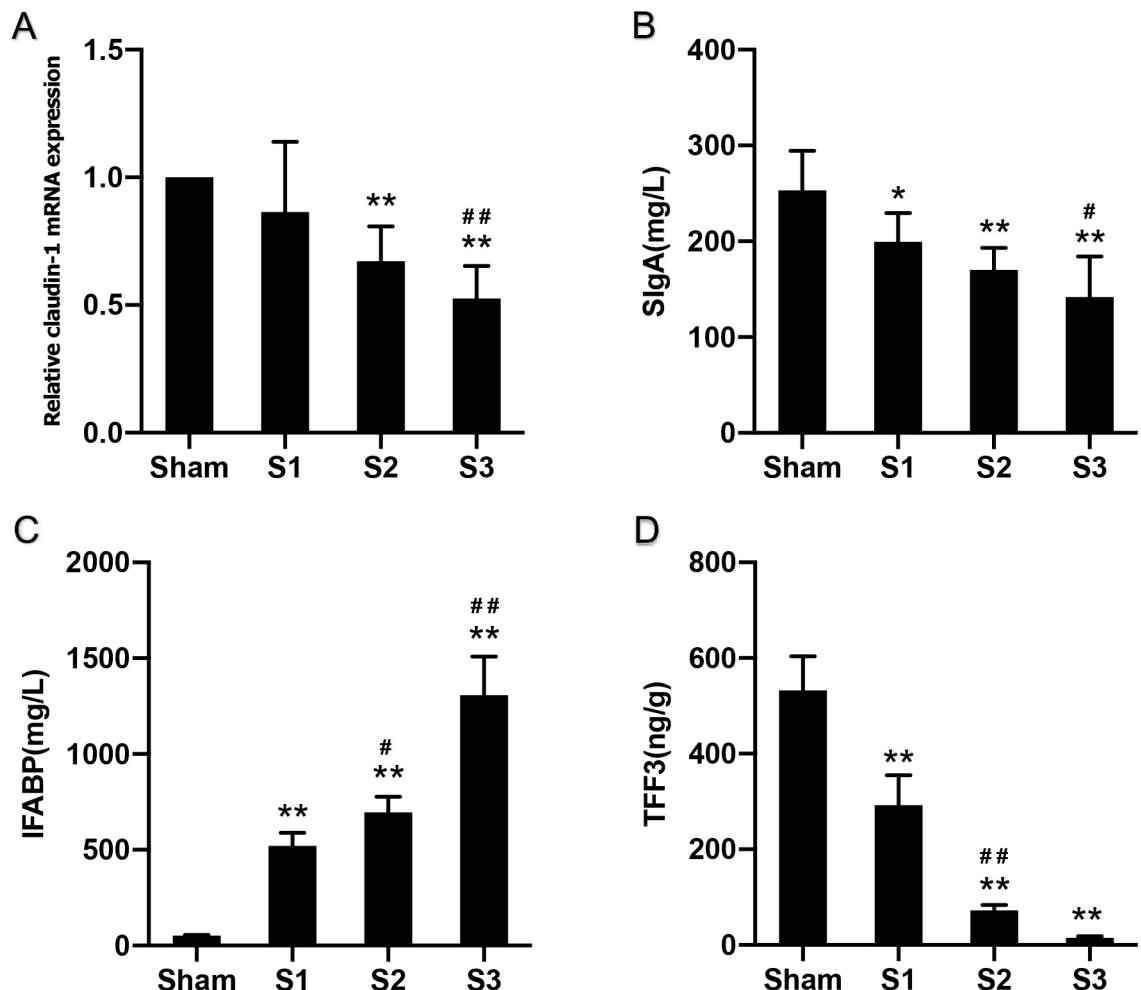

**Fig 11. Changes in indicators of intestinal mucosal barrier function.** Data are presented as the mean ± SD, n = 5. * P<0.05, ** P<0.01 vs. Sham group; #P<0.05, ## P<0.01 vs. the preceding group.

Recent clinical studies have found that the reduction of intestinal microbiota diversity is associated with many human diseases [34]. Loss of microbiota diversity appears as a common feature not only in digestive diseases such as Crohn's disease [35], irritable bowel syndrome

**Table 3. Correlations between intestinal flora and indicators of intestinal mucosa.**

| Intestinal flora | Correlation with claudin-1 | | Correlation with sIgA | | Correlation with IFABP | | Correlation with TFF3 | |
|---|---|---|---|---|---|---|---|---|
| | r | P-value | r | P-value | r | P-value | r | P-value |
| *Romboutsia* spp. | 0.519 | 0.048* | 0.683 | <0.001** | -0.680 | 0.001** | 0.811 | <0.001** |
| *Turicibacter* spp. | 0.420 | 0.119 | 0.632 | 0.001** | -0.627 | 0.003** | 0.679 | 0.001** |
| *Lactobacillus* spp. | 0.463 | 0.082 | 0.335 | 0.110 | -0.434 | 0.056 | 0.500 | 0.025* |
| *Akkermansia* spp. | -0.534 | 0.040* | -0.639 | 0.001** | 0.318 | 0.172 | -0.441 | 0.052 |
| *Escherichia-Shigella* | -0.540 | 0.038* | -0.607 | 0.002** | 0.214 | 0.364 | -0.374 | 0.104 |
| *Bacteroides* spp. | -0.596 | 0.019* | -0.705 | <0.001** | 0.642 | 0.002** | -0.721 | <0.001** |

*P<0.05

**P<0.01.

[36] and colorectal cancer [37] but also in non-digestive diseases such as diabetes, obesity and autism. However, in the sequencing data, we found that the bacterial diversity of the gut microbiota was significantly greater in SBO rats compared with sham-operated rats and continuously increased with prolonged obstruction duration, indicating higher species richness and greater evenness in blocked intestine. This finding is consistent with the results in the model of partial colon obstruction [22]. Besides, it is interesting to note that samples got closer to sham-operated group with prolonged obstruction duration in PCA and PCoA results, which could be caused by an increase abundance of bacterial genera belonging to Firmicutes.

In addition, comparative analysis of bacterial community composition identified several differentially abundant bacterial groups in SBO rats. The relative abundance of bacterial genera that are generally considered to be associated with gut health like *Romboutsia*, *Turicibacter*, *Lactobacillus* and *Bifidobacterium* were decreased in rats with SBO, whereas certain bacteria like *Akkermansia*, *Escherichia-Shigella*, *Bacteroides*, *Enterobacter* and *Lysinibacillus* were increased. In the early stages of obstruction, *Akkermansia* and *Escherichia-Shigella* were the dominant bacterial genus. Although many reports showed that *Akkermansia* is attracted to signs of human health as its abundance is inversely correlated with different type of diseases [38], Reunanen *et al.* reported that high doses of *Akkermansia muciniphila* could cause the increase of IL-8 release and enhanced proinflammatory activity in the epithelium [39]. Similar to the findings of Reunanen *et al.*, the abundance of *Akkermansia* showed the same trends as *Escherichia-Shigella* in SBO presenting a mucosa-damaging effect to intestinal mucosa barrier. *Escherichia-Shigella* belonging to the phylum Proteobacteria can produce LPS, also called endotoxins, which is released in the process of normal metabolism or after destruction of the bacterial cell wall [40]. LPS are large molecules and normally cannot penetrate the bowel wall; however, high LPS concentrations on the bowel wall can lead to the disruption of tight junction barrier and intestinal permeability [41]. Leakage of the LPS into the blood causes serious endotoxemia and systemic inflammatory response syndrome in SBO. The results in our research suggest that *Akkermansia* and *Escherichia-Shigella* may have a synergistic effect on damaging intestinal mucosal barrier in the process of SBO. With the prolongation of obstruction time, the relative abundance of *Bacteroides* increased and was equivalent to *Akkermansia* or *Escherichia-Shigella*. These three bacteria became the dominant bacterial genus at a late stage of obstruction. The immense number of *Escherichia-Shigella* entails colonization resistance for other bacteria especially aerobes. *Bacteroides*, obligate anaerobic bacteria, is untouched and multiply quickly causing a feculent odor of the intestinal contents [42].

Sequencing consequences provided the features of entire microbial communities, and then we determined the absolute quantity of target bacterial groups by qPCR. The copy numbers of 16S rRNA gene of total bacteria in rats with small bowel obstruction increased significantly at 24 h, but no statistically significant difference was observed from 24 to 72 hours. Growing characteristics of total bacteria showed an exponential growth within 24 hours and entered a stationary phase afterward. This stationary phase may be due to several growth-limiting factors such as the impairment of intestinal mucus layer, the consumption of an essential nutrient, and competition between bacterial species. Studies have shown that *E. coli* usually reaches its maximum concentration within a few hours, and no later than 24 hours in animals or human subjects with small bowel obstruction [15, 16]. This is in consistent with our findings and we also noted parallel changes in *Akkermansia*. Although the relative abundance of *Escherichia-Shigella* and *Akkermansia* showed decreasing trends with prolonged obstruction time, the absolute bacterial cell numbers had no significant change. This result indicated that the number of microbial species was increasing and more unknown species appeared, which led to declining percentages of those bacterial genera.

Under normal conditions, the efficiency of intestinal barrier function depends on a complex network of cellular, immunological, biochemical and microbial factors [43], such as intestinal epithelial cells, tight junction proteins, intraepithelial lymphocytes, immunoglobulin A, mucus layer or symbiotic bacteria [44]. Claudin-1 is a protein belonging to tight junctions between intestinal epithelial cells and an important regulator of epithelial barrier function. Secretory IgA (sIgA) is an immunoglobulin produced by mucous membranes and promotes humoral immunity at the intestinal mucosa. Therefore, claudin-1 and sIgA could act as appropriate biomarkers of homeostasis or dysfunction of the gut barrier. In the present study, the levels of claudin-1 gene expression and sIgA decreased as prolonged obstruction presenting a serious impairment of the intestinal mucosal barrier due to bowel obstruction.

Intestinal fatty acid binding protein (IFABP) is a cytosolic protein that play an important role in the cellular uptake and is expressed by enterocytes of the small intestine. It is released when gastrointestinal mucosal integrity is destroyed and mucosal tissue injury. Many researches have suggested that IFABP might be a useful biochemical marker for acute intestinal ischaemia and inflammatory bowel damage [45, 46]. Trefoil factor 3 (TFF3) is a secreted glycoprotein produced by goblet cells and also plays a key role in mucosal protection. Studies have showed that TFF3 was a regulator of many gastrointestinal diseases such Ulcerative colitis and Crohn's disease [47]. In this study, intestinal mucosal injury has been observed to be attended by a significant increase in IFABP and a reduction of TFF3 levels.

Results from the Spearman's correlation analysis showed positive correlations of sIgA with *Romboutsia* and *Turicibacter*. These two bacterial genera are found to be abundant inhabitants of small intestine in rats and are generally considered beneficial to the hosts [48, 49]. Meanwhile, our data revealed that mucosa-damaging bacteria (*Akkermansia*, *Escherichia-Shigella* and *Bacteroides*) negatively correlated with the claudin-1 and sIgA in SBO rats. Although previous studies showed that *Akkermansia muciniphila* modulates pathways involved in intestinal integrity, basal metabolism and intestinal immunity [50], our results indicated *Akkermansia* may reduce the gene expression of the tight junction protein and the level of immunoglobulin resulting in increased intestinal permeability and weakened immune system, which was consistent with the results of sequencing data. *Escherichia-Shigella* and *Bacteroides* aggravated the impairment of the gut barrier and bacterial translocation by producing LPS [51, 52].

Although gut bacteria showed a significant change in small bowel obstruction, many other factors could influence the composition of the microbiota in rat models, such as genetics, housing, diet, experimental artifact and host immune system [53]. Thus, we housed animals in independent cages to reduce the rate of natural microbial drift, and provided autoclaved chow and sterilized water to reduce the interference of external factors. However, the effect of experimental artifact in this study is inevitable. We need to find a method to create complete small bowel obstruction models without invading the abdominal cavity in the future. Besides, the animals will undergo a series of pathophysiological changes with prolonged obstruction, such as loss of body fluid and electrolyte, disturbance of acid-base balance, infection and poisoning, etc. These systemic changes can affect the changes of intestinal flora, and the disturbance of flora will further aggravate the body damage, forming a vicious circle. This experimental result is a supplement to the pathophysiological data of small bowel obstruction and provides a reference for clinical exploration of the changes in intestinal microbiota of SBO. Considering that this study only provided short-term alteration in the gut microbiota during SBO, long-term experiments should be conducted in the following study to explore sustained changes in gut microbiota and mucosal barrier.

In summary, small bowel obstruction led to structural and quantitative alterations of gut microbiota. SBO decreased relative abundance of Firmicutes, especially *Romboutsia* and *Turicibacter*, and increased abundance of Proteobacteria, Verrucomicrobia and Bacteroidetes.

Obstruction also significantly increased microbiota diversity in small bowel and the absolute quantity of total bacteria within 24 hours. However, the total bacterial count did not change significantly from 24 to 72 hours while the number of microbial species still increased. Disruption of intestinal mucosal barrier appeared in SBO with decreases in the expression of claudin-1 mRNA, sIgA and TTF3 levels. *Akkermansia*, *Escherichia-Shigella* and *Bacteroides* may have synergistic effects on damaging gut barrier and immune system in SBO.

## Supporting information

**S1 File. Data of OTUs used for gut microbial analysis.**
(FASTA)

**S1 Table. Count number of OTUs by taxonomic rank for each sample.**
(XLS)

**S1 Dataset. Data of qPCR analysis of target bacterial groups.**
(XLSX)

**S2 Dataset. Data of ELISA results in the different experimental groups.**
(XLSX)

## Acknowledgments

We greatly thank Teng Wu and Jinjin Liu of the department of Pharmacology for their assistance with animal experiment.

## Author Contributions

**Conceptualization:** Nan Zhang, Jiliang Xie.

**Data curation:** Jiali Mo, Liuyang Fan.

**Formal analysis:** Jiali Mo.

**Investigation:** Jiali Mo.

**Methodology:** Jiali Mo, Donghua Li, Tao Shan, Liuyang Fan.

**Resources:** Donghua Li.

**Supervision:** Jiliang Xie.

**Writing – original draft:** Jiali Mo.

**Writing – review & editing:** Lei Gao, Nan Zhang.

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
