## [Decision Letter · Decision Letter 0]

10 Nov 2020

PONE-D-20-27802

Structural and quantitative alterations of gut microbiota in experimental small bowel obstruction

PLOS ONE

Dear Dr. Zhang,

Thank you for submitting your manuscript to PLOS ONE. After careful consideration, we feel that it has merit but does not fully meet PLOS ONE’s publication criteria as it currently stands. Therefore, we invite you to submit a revised version of the manuscript that addresses the points raised during the review process.

We look forward to receiving your revised manuscript.

Kind regards,

Christopher Staley, Ph.D.

Academic Editor

PLOS ONE

Journal Requirements:

2. Please describe the training that researchers had to kill rats by cervical dislocation.

3. Please provide the name and catalog number of the ELISA kit used.

4. Please change p-values of 0.000 in your tables to <0.001.

Additional Editor Comments:

The authors' manuscript has been reviewed by myself and two experts. As the reviewers note, clarification of several methodological details is required. In addition, it is suggested the authors expand on the potential clinical relevance of their results.

Reviewers' comments:

Reviewer's Responses to Questions

**Comments to the Author**

1. Is the manuscript technically sound, and do the data support the conclusions?

Reviewer #1: Partly

Reviewer #2: Partly

2. Has the statistical analysis been performed appropriately and rigorously? 

Reviewer #1: Yes

Reviewer #2: Yes

3. Have the authors made all data underlying the findings in their manuscript fully available?

Reviewer #1: Yes

Reviewer #2: Yes

4. Is the manuscript presented in an intelligible fashion and written in standard English?

Reviewer #1: No

Reviewer #2: Yes

5. Review Comments to the Author

Reviewer #1: This study characterized the gut microbiota changes in a surgically constructed small bowel obstruction rat model. The authors observed the post-surgical change of relative and absolute abundance of different microbial species. In addition, the author observed increased expression of pro0inflammatory markers including claudin-1 mRNA and serum IgA. Although it is the first time that the microbiota shifts have been reported in this experimental rat model, not much insights were drawn from these observations. Moreover, the clinical relevance of the rat model or the observation is unclear. Thus, revisions would be necessary to draw scientific or clinical insights of these observations reported.

Some specific concerns:

The study reported the change of different microbial species after surgically placing a polyvinyl chloride clip around the terminal ileum. The change of microbiota, per se, is expected and not surprising. As we know, the microbiota is shaped by environmental factors including nutrient, oxygen level, P.H., host inflammation, etc. The surgery would have caused turbulence in all these factors, and as a result, it is inevitable that the microbiota also changes. The important questions here are whether these changes have any impact on the progress or recovery of the disease, and which of the environmental factors play the most important role in reshaping microbiota. These questions remain unanswered in the manuscript.

In addition, the authors need to address how the observations in this study translate to clinical situations. Are similar changes observed in clinical studies of small bowel observation? Without such information, the changes of microbiota in this study could very well be attributed to experimental artifact caused by the physical clipping. Alternatively, more biomarkers on the host side could address the clinical relevance of such an experimental setup. Serum IgA and claudin A are not SBO specific markers; they are rather general pro-inflammatory markers. The elevated level could be due to the mucosal damage caused by the surgical processes. Some more specific host markers should be used.

Technical concern:

Bedding and cage effects are important factors to control for microbiome study. The authors need to show that prior to the surgery, all the four groups have similar microbiota. This could be done by including a PCoA analysis of prior surgical fecal sample. In addition, the authors should include more details on mix-bedding and cage arrangements of the cohorts.

Minor points:

In the introduction, there are bold claims without evidence. For example, Line 45: “Gut microbiota dysbiosis is a prominent cause of both local and systemic pathological changes in SBO.” And line 50: “Microbiota dysbiosis is considered a major pathogenic factor in small bowel obstruction.” I don’t think the cited articles or the current study can prove such bold claims of causality between microbiota composition and SBO. If the statements were true, then the authors didn’t need to use surgical clips to create a rat SBO model.

Some sentences in the introduction and discussion sessions are unnecessarily long and cause confusion. For example, line 60: “Although sequencing consequences of entire microbial communities are very informative, they may not be able to capture the essence of the problem, which means that it is the absolute quantity and not the relative abundance of microbes that really matters.” This long sentence needs rewriting as the message is not clear.

Please define all the abbreviations. For example, does sIgA stand for serum IgA or secretary IgA in this context?

Reviewer #2: This study examines the microbiota change in rats with complete obstruction. Not surprisingly, there changes that evolved over time in both the tissue and the bacteria. Several points need to be addressed.

1. The rat model should be described in more detail as the references given are different from these rats. What gross changes happened in these rats - weight, intestinal diameter, food and water intake, etc.?

2. The quantify the total bacterial content, it is necessary to consider the intestinal dilation that occurs with bowel obstruction. The enteric content must have increased with obstruction. The total volume of the content should be given in these rats and considered in the calculations.

3. With prolonged obstruction, these rats must be getting significantly dehydrated - how is that taken into account in the data interpretation? Could the observed changes occur due to dehydration rather than obstruction?

4. What is the food intake during the experiments? That might also impact the results of the study.

6. PLOS authors have the option to publish the peer review history of their article (what does this mean?). If published, this will include your full peer review and any attached files.

Reviewer #1: No

Reviewer #2: No

---

## [Author Response · Author response to Decision Letter 0]

28 Jan 2021

Replies to Reviewer 1

Q: The surgery would have caused turbulence in all these factors, and as a result, it is inevitable that the microbiota also changes. The important questions here are whether these changes have any impact on the progress or recovery of the disease, and which of the environmental factors play the most important role in reshaping microbiota. These questions remain unanswered in the manuscript.

A: The surgery will have impact on intestinal microbiota indeed, but the obstruction ring in this study can cause less damage to the intestinal wall than ligation with silk suture. We will try our best to create a complete small bowel obstruction model without invading the abdominal cavity in the future. Besides, we set up a control group to reduce the influence of surgical operations. The environmental factors have been discussed in the section of discussion.

Q: Serum IgA and claudin A are not SBO specific markers; they are rather general pro-inflammatory markers. Some more specific host markers should be used.

A: We have added intestinal fatty acid binding protein (IFABP) and Trefoil factor 3 (TFF3) as specific host markers.

Q: A PCoA analysis of prior surgical fecal sample. In addition, the authors should include more details on mix-bedding and cage arrangements of the cohorts.

A: We have added PCoA analysis of prior surgical fecal sample in the section of results. More details of mix-bedding and cage arrangements have added in the section of methods.

Q: In the introduction, there are bold claims without evidence. Some sentences in the introduction and discussion sessions are unnecessarily long and cause confusion.

A: We have changed those expressions without evidence or causing confusion.

Q: Please define all the abbreviations.

A: We have defined all the abbreviations.

Replies to Reviewer 2

Q: The rat model should be described in more detail as the references given are different from these rats. What gross changes happened in these rats - weight, intestinal diameter, food and water intake, etc.?

A: We have detailed the rat model and general condition in the sections of methods and results.

Q: The quantify the total bacterial content, it is necessary to consider the intestinal dilation that occurs with bowel obstruction. The enteric content must have increased with obstruction. The total volume of the content should be given in these rats and considered in the calculations.

A: The copy numbers of target bacterial 16S rRNA genes were calculated per milligram of sample, so the total volume of content might be not that important.We will take it into consideration in the following experiment. 

Q: With prolonged obstruction, these rats must be getting significantly dehydrated - how is that taken into account in the data interpretation? Could the observed changes occur due to dehydration rather than obstruction?

A: Dehydration would lead to local and systemic pathophysiological changes in animals. There was a large amount of gas-liquid accumulation in the intestine when obstructed, so we thought that dehydration had little impact on the results of bacterial sequencing. It may have an influence on the serum indicators.

Q: What is the food intake during the experiments? That might also impact the results of the study.

A: Rats of SBO groups reduced food intake in 24 hours and stopped taking food gradually after that. We provided autoclaved chow and sterilized water to reduce the interference of external factors.

---

## [Decision Letter · Decision Letter 1]

4 Mar 2021

PONE-D-20-27802R1

Structural and quantitative alterations of gut microbiota in experimental small bowel obstruction

PLOS ONE

Dear Dr. Zhang,

Thank you for submitting your manuscript to PLOS ONE. After careful consideration, we feel that it has merit but does not fully meet PLOS ONE’s publication criteria as it currently stands. Therefore, we invite you to submit a revised version of the manuscript that addresses the points raised during the review process.

We look forward to receiving your revised manuscript.

Kind regards,

Christopher Staley, Ph.D.

Academic Editor

PLOS ONE

Reviewers' comments:

Reviewer's Responses to Questions

**Comments to the Author**

1. If the authors have adequately addressed your comments raised in a previous round of review and you feel that this manuscript is now acceptable for publication, you may indicate that here to bypass the “Comments to the Author” section, enter your conflict of interest statement in the “Confidential to Editor” section, and submit your "Accept" recommendation.

Reviewer #1: All comments have been addressed

Reviewer #2: (No Response)

2. Is the manuscript technically sound, and do the data support the conclusions?

Reviewer #1: Yes

Reviewer #2: No

3. Has the statistical analysis been performed appropriately and rigorously? 

Reviewer #1: Yes

Reviewer #2: No

4. Have the authors made all data underlying the findings in their manuscript fully available?

Reviewer #1: Yes

Reviewer #2: No

5. Is the manuscript presented in an intelligible fashion and written in standard English?

Reviewer #1: Yes

Reviewer #2: Yes

6. Review Comments to the Author

Reviewer #1: The authors have addressed all my comments in the previous review. The new data and figures are appropriate. I don't have any more questions.

Reviewer #2: In the revised manuscript, the authors did not give adequate responses to the previous reviews. Many confounding factors such as altered food intake, hydration status, and enteric volume make the studies hard to interpret.

7. PLOS authors have the option to publish the peer review history of their article (what does this mean?). If published, this will include your full peer review and any attached files.

Reviewer #1: No

Reviewer #2: No

---

## [Author Response · Author response to Decision Letter 1]

25 Apr 2021

Replies to Reviewer 2

Q: What gross changes happened in these rats - weight, intestinal diameter, food and water intake, etc.?

A: We have detailed the general condition in the sections of results.

Q: The total volume of the content should be given in these rats and considered in the calculations.

A: We have added the total volume of the content into calculation and the results were presented in Fig 10.

Q: With prolonged obstruction, these rats must be getting significantly dehydrated - how is that taken into account in the data interpretation? 

A: We have added some discussions at the end of essay.

---

## [Decision Letter · Decision Letter 2]

18 May 2021

PONE-D-20-27802R2

Structural and quantitative alterations of gut microbiota in experimental small bowel obstruction

PLOS ONE

Dear Dr. Zhang,

Thank you for submitting your manuscript to PLOS ONE. After careful consideration, we feel that it has merit but does not fully meet PLOS ONE’s publication criteria as it currently stands. Therefore, we invite you to submit a revised version of the manuscript that addresses the points raised during the review process.

Additional minor clarifications are requested, as noted by the reviewer.

We look forward to receiving your revised manuscript.

Kind regards,

Christopher Staley, Ph.D.

Academic Editor

PLOS ONE

Journal Requirements:

Reviewers' comments:

Reviewer's Responses to Questions

**Comments to the Author**

1. If the authors have adequately addressed your comments raised in a previous round of review and you feel that this manuscript is now acceptable for publication, you may indicate that here to bypass the “Comments to the Author” section, enter your conflict of interest statement in the “Confidential to Editor” section, and submit your "Accept" recommendation.

Reviewer #3: (No Response)

2. Is the manuscript technically sound, and do the data support the conclusions?

Reviewer #3: Yes

3. Has the statistical analysis been performed appropriately and rigorously? 

Reviewer #3: Yes

4. Have the authors made all data underlying the findings in their manuscript fully available?

Reviewer #3: Yes

5. Is the manuscript presented in an intelligible fashion and written in standard English?

Reviewer #3: Yes

6. Review Comments to the Author

Reviewer #3: This is an interesting manuscript on the impact of small bowel obstruction on the composition of the gut microbiota and its potential impacts on the gut barrier function.

The methodology is clear, and the results are well presented. The findings provide an interesting and novel overview of changes in the gut microbiota in the days following SBO.

The authors have addressed to the best of their ability the comments raised by previous reviewers and made many improvements to the manuscript.

Nonetheless, several issues require attention.

1. In the introduction, the mortality rate associated with SBO (reported to be 13%) seems high compared to the literature. Other sources report much lower rates, and this value should be nuanced (suggested reference: Paulson EK, Thompson WM. Review of small-bowel obstruction: the diagnosis and when to worry. Radiology. 2015 May;275(2):332-42. doi: 10.1148/radiol.15131519. PMID: 25906301).

2. Since SBO may not be understandable to all readers, I would rephrase the following sentence (line 52): “Microbiota dysbiosis is considered a major pathophysiological process in small bowel obstruction” to make clearer the fact that such dysbiosis is a consequence of the SBO process.

3. It is mentioned that the after surgery, the content of the gut above the obstruction point was collected for analysis. Was the content of the same segment collected in rats in the control (sham) group? This is important to specify since the comparison with control rats would be impossible otherwise.

4. Figure 6 - the treatment groups should be specified (sham, S1, S2, S3).

5. Overall, it is expected that a SBO is an acute event that would alter the gut microbiota composition, but the relevant conclusion from this work is a potential deleterious and durable effect that may affect the host’s health beyond the few days where SBO is present. I understand that this is not the purpose of the manuscript, but I would discuss the necessity of conducting long-term experiments in the discussion to investigate possible sustained “dysbiosis” and/or barrier dysfunction.

6. The following sentence needs attention: “A small loop of ileum with mesentery approximately 1 cm distal to the ileocecal valve”. The ileum is proximal to the ileocecal valve anatomically and the word “distal” is inaccurate should be corrected.

7. The following sentence requires attention as well: “A 2 cm long ileal segment 2 cm oral to the obstruction”. Does the word “oral” means proximal or cranial? It should be modified for clarity.

8. In the discussion, the authors suggest that the findings may orient the choice of antibiotics in SBO. However, SBO is not an infectious phenomenon, and no antibiotics are administered routinely, unless a complication (ischemia or perforation) occurs, in which case the used antibiotics are large spectrum already.

9. The main limitation of this study is the message it conveys. The findings are interesting and are worth sharing with the scientific community. These findings however should be seen as the short-term alteration in the gut microbiota during SBO. They do not provide however information on long-term alterations or health risks that SBO may induce. This is especially true as SBO does not seem in practice to induce any long-term diseases in patients. In the discussion, the message should be centered on the effect of SBO on gut microbiota in an acute setting only, as long-term alterations cannot be described or even extrapolated with the present study.

7. PLOS authors have the option to publish the peer review history of their article (what does this mean?). If published, this will include your full peer review and any attached files.

Reviewer #3: No

---

## [Author Response · Author response to Decision Letter 2]

8 Jul 2021

Replies to Reviewer 3

Q: In the introduction, the mortality rate associated with SBO (reported to be 13%) seems high compared to the literature. Other sources report much lower rates, and this value should be nuanced.

A: We have corrected the mortality rate and added the reference.

Q: Since SBO may not be understandable to all readers, I would rephrase the following sentence (line 52): “Microbiota dysbiosis is considered a major pathophysiological process in small bowel obstruction” to make clearer the fact that such dysbiosis is a consequence of the SBO process.

A: We have changed the expression in line 52.

Q: It is mentioned that the after surgery, the content of the gut above the obstruction point was collected for analysis. Was the content of the same segment collected in rats in the control (sham) group?

A: We have added the explanation in line 101.

Q: Figure 6 - the treatment groups should be specified (sham, S1, S2, S3).

A: We have added the groups in figure 6.

Q: Discuss the necessity of conducting long-term experiments in the discussion to investigate possible sustained “dysbiosis” and/or barrier dysfunction.

A: We have added the discussion in line 437.

Q: The ileum is proximal to the ileocecal valve anatomically and the word “distal” is inaccurate should be corrected.

A: We have corrected the word in line 88.

Q: The following sentence requires attention as well: “A 2 cm long ileal segment 2 cm oral to the obstruction”. Does the word “oral” means proximal or cranial? It should be modified for clarity.

A: We have corrected the word in line 115.

Q: In the discussion, the authors suggest that the findings may orient the choice of antibiotics in SBO. However, SBO is not an infectious phenomenon, and no antibiotics are administered routinely, unless a complication (ischemia or perforation) occurs, in which case the used antibiotics are large spectrum already.

A: We have deleted that expression.

Q: In the discussion, the message should be centered on the effect of SBO on gut microbiota in an acute setting only, as long-term alterations cannot be described or even extrapolated with the present study.

A: We have focused on the acute changes in microbiota in SBO and deleted discussion about long-term changes.

---

## [Decision Letter · Decision Letter 3]

22 Jul 2021

Structural and quantitative alterations of gut microbiota in experimental small bowel obstruction

PONE-D-20-27802R3

Dear Dr. Zhang,

We’re pleased to inform you that your manuscript has been judged scientifically suitable for publication and will be formally accepted for publication once it meets all outstanding technical requirements.

Kind regards,

Christopher Staley, Ph.D.

Academic Editor

PLOS ONE

Additional Editor Comments (optional):

Reviewers' comments:

Reviewer's Responses to Questions

**Comments to the Author**

1. If the authors have adequately addressed your comments raised in a previous round of review and you feel that this manuscript is now acceptable for publication, you may indicate that here to bypass the “Comments to the Author” section, enter your conflict of interest statement in the “Confidential to Editor” section, and submit your "Accept" recommendation.

Reviewer #3: All comments have been addressed

2. Is the manuscript technically sound, and do the data support the conclusions?

Reviewer #3: Yes

3. Has the statistical analysis been performed appropriately and rigorously? 

Reviewer #3: Yes

4. Have the authors made all data underlying the findings in their manuscript fully available?

Reviewer #3: Yes

5. Is the manuscript presented in an intelligible fashion and written in standard English?

Reviewer #3: Yes

6. Review Comments to the Author

Reviewer #3: The raised comments were addressed by the authors. The discussion was also nuanced to account for the limitations of the study. The present version of the manuscript seems suitable for publication in its present form.

7. PLOS authors have the option to publish the peer review history of their article (what does this mean?). If published, this will include your full peer review and any attached files.

Reviewer #3: **Yes: **Manuela Santos

---

## [Editor Report · Acceptance letter]

26 Jul 2021

PONE-D-20-27802R3 

Structural and quantitative alterations of gut microbiota in experimental small bowel obstruction 

Dear Dr. Zhang:

I'm pleased to inform you that your manuscript has been deemed suitable for publication in PLOS ONE. Congratulations! Your manuscript is now with our production department. 

Kind regards, 

on behalf of

Dr. Christopher Staley 

Academic Editor

PLOS ONE